# Resilience of hospital and allied infrastructure during pandemic and post pandemic periods for maternal health care of pregnant women and infants in Tamil Nadu, India - A counterfactual analysis

**Kandaswamy Paramasivan**[1]*, **Ashwin Prakash**[2], **Sarthak Gupta**[2], **Bhairav Phukan**[2], **Pavithra M.R.**[3], **Balaji Venugopal**[4,5]

1 Department of Management Studies, Indian Institute of Technology, Madras, Chennai, India, 2 Department of Computer Science, SRM Institute of Science and Technology, Chengalpattu, Tamil Nadu, India, 3 Great Lakes Institute of Management, Chennai, India, 4 University of Glasgow, Glasgow, United Kingdom, 5 Beatson West of Scotland Cancer Center, Glasgow, United Kingdom

* kandy@berkeley.edu

## Abstract

COVID-19 has impacted the healthcare system across the globe. The study will span three pandemic waves in 2020, 2021, and 2022. The goal is to learn how the pandemic affects antenatal care (ANC) and emergency delivery care for pregnant women in Tamil Nadu, India, and how medical services respond. The study employs counterfactual analysis to evaluate the causal impact of the pandemic. A feedforward in combination with a simple auto-regressive neural network (AR-Net) is used to predict the daily number of calls for ambulance services (CAS). Three categories of the daily CAS count between January 2016 and December 2022 are utilised. The total CAS includes all types of medical emergencies; the second group pertains to planned ANC for high-risk pregnant women and the third group comprises CAS from pregnant women for medical emergencies. The second wave's infection and mortality rates were up to six times higher than the first. The phases in wave-II, post-wave-II, wave-III, and post-wave-III experienced a significant increase in both total IFT (inter-facility transfer) and total non-IFT calls covering all emergencies relative to the counterfactual, as evidenced by reported effect sizes of 1 and a range of 0.65 to 0.85, respectively. This highlights overwhelmed health services. In Tamil Nadu, neither emergency prenatal care nor planned prenatal care was affected by the pandemic. In contrast, the increase in actual emergency-related IFT calls during wave-II, post-wave-II, wave-III, and post-wave-III was 62%, 160%, 141%, and 165%, respectively, relative to the counterfactual. During the same time periods, the mean daily CAS related to prenatal care increased by 47%, 51%, 38%, and 38%, respectively, compared to pre-pandemic levels. The expansion of ambulance services and increased awareness of these services during wave II and the ensuing phases of Covid-19 pandemic have enhanced emergency care delivery for all, including obstetric and neonatal cohorts.

**Data Availability Statement:** All relevant data are within the paper and its supporting information files

**Funding:** The author(s) received no specific funding for this work

**Competing interests:** The authors have declared that no competing interests exist

## Introduction

### Global trend in maternal health care during pandemics

The COVID-19 pandemic has caused severe disruptions in healthcare utilisation and delivery across the globe [1, 2]. The healthcare system in every nation is still striving to strike a delicate balance between responding to the pandemic crisis and continuing to provide other essential services [3]. A pre-existing organized, financed, and well-coordinated health care system is prudent for essential services to remain available in a pandemic situation [4]. In low and medium-income countries, such essential services include screening and treatment of communicable diseases (e.g., tuberculosis and human immuno deficiency virus[HIV]), maternal and child healthcare (MCH), treatment of injuries and surgical conditions, and cancer screenings [5]. Delivery of these services mainly affects vulnerable cohorts such as women and children. One of the most serious concerns about this pandemic was its impact on antenatal and postnatal care, as well as pregnant women's access to in-person clinics [6].

During the COVID-19 pandemic, global maternal and foetal outcomes have deteriorated with an increase in maternal deaths, stillbirths, ruptured ectopic pregnancies, and maternal depression. Certain outcomes demonstrate substantial disparities between high-resource and low-resource environments [7, 8]. Maternal deaths rose in the United States during the pandemic years of 2020 and 2021 compared to 2018 and 2019. Importantly, 25% of all maternal deaths in 2020 and 2021 were caused by COVID-19 [9]. Slomski and colleagues reported that the maternal death rate rose to 25.1 per 100,000 live births from April to December 2020 up 33.3% from the previous quarter. The study period saw increases in the maternal mortality rates of 17.2%, 40.2% and 74.2% respectively, in non-Hispanic whites, non-Hispanic black and Hispanics [10]. The study of how the pandemic has affected maternal health care in Haiti, Lesotho, Liberia, Malawi, Mexico, and Sierra Leone reported a considerable decline in the number of first antenatal care visits in Haiti (18%) and Sierra Leone (32%), and facility-based births in all countries except Malawi from March to December 2020 [11]. The systematic review examined the utilisation of maternal and child healthcare (MCH) services during pandemics (Zika, Ebola, and COVID-19) and the effectiveness of various interventions undertaken to ensure MCH care service utilisation. The review indicates that MCH utilisation is frformuently affected during pandemics [12]. Equally, the impact of the pandemic on MCH services in selected districts of Assam was the subject of a more pertinent study concerning India. It revealed that healthcare providers cited a lack of transportation facilities and a lack of pharmaceutical supply as barriers to providing routine services [13].

### Maternal and child health care (MCH) set up in India

In India, multiple MCH programmes have been successfully implemented to provide universal health care to pregnant women and minimise the maternal mortality. The southern state of Tamil Nadu is renowned for having successfully implemented these schemes and has lower maternal mortality rates. Of these, a targeted approach to promote ambulance use among antenatal mothers to transport pregnant women to an appropriate health facility is one of the significant schemes that play a pivotal role in reducing maternal mortality [14].

The federal and state governments have recently worked to eliminate all maternal and infant deaths through the delivery of high-quality, respectful, and dignified care through a variety of programmes. Due to government led initiatives in India, the maternal mortality ratio (MMR) decreased by 80% from 556 per 100,000 live births in 1990 to 113 per 100,000 live births in 2016–18 compared to the global decline of 45%. In addition, the MMR decreased by

8.8% between 2016–17 and 2017–18 from 113 to 103 per 100,000 live births. These national programmes aimed to provide high-quality obstetric and new-born care to every pregnant woman and unborn child who visited a public health facility [15]. Currently there are 7 states that have met the sustainable development goal target of MMR of 70 per 100,000 live births, in which Tamil Nadu has achieved 58 per 100,000 live births [16].

In response, on October 10, 2019, the Indian government unveiled the comprehensive "Surakshit Matritva Aashwasan" (SUMAN) programme with an objective of consolidating the benefits of multiple maternal health-related initiatives, including "Ayushman Bharat," "Indian Public Health Services" and "LaQshya". Beneficiaries of SUMAN include expectant mothers, mothers of infants under six months old, sick infants, and children under one year. The programme establishes a call centre for better redressal of complaints using cutting-edge technology.

For pregnant women designated as beneficiaries, a minimum of four antenatal visits are necessary. These women receive prompt diagnostic care and suitable medical treatments thanks to the state's infrastructure network of government hospitals. These targeted women receive notice of the consultation date in advance, which is normally a Tuesday, and use ambulance services to travel to the consultation facility, which is typically one of the district-level government hospitals. The implementation of SUMAN is overseen by four committees, one for each level of government (national, state, district, and block). The SUMAN initiative, which aims to give expectant mothers and mothers of young children better maternal care, is carried out by district-level and state-level committees. In contrast, the national committee that advises the government meets twice a year. According to the National Health Systems Resource Centre India report, as of October 21, 2021, Tamil Nadu ranked first in India with 644 basic emergency obstetric and new-born care (BEmONC) and ranked second in India with 126 comprehensive emergency obstetric and new-born care (CEmONC) [17]. Globally many non-emergency medical procedures were postponed, and health care providers were focusing on care of the COVID-19 pandemic-related illness [18, 19].

## Impact of pandemic in India

The pandemic (2020–2023) caused significant pandemonium and danger worldwide. The way it repeated itself suggested that it would eventually engulf most of the human population, both spatially and temporally, in successive waves. As nearly every nation was affected, it posed a grave threat to the public and governments, as countries with more advanced and equipped medical infrastructure and sufficient supplies and services were unable to assist other nations. It is alarming to learn how many people were hospitalised and died from the disease based on the data gathered during the three pandemic waves in India. Even a low mortality rate will have a significant impact on such a populous nation. The second wave of COVID-19 (2021) had catastrophic infection rates. Due to the pandemic, India experienced acute shortages of medical supplies, emergency equipment, intensive care unit (ICU) beds, and oxygen cylinders [20, 21]. The medical and paramedical teams involved in handling the situation found it to be a great challenge. Despite the fact that wave-III's infection rate in 2022 was not lower, it was mild and had very few fatalities.

Contrary to other medical care and treatment, the situation of pregnant women is unique because prompt interventions are essential for the health of the mother and the unborn child. During the peak of the second wave in 2021, the pandemic derailed numerous health projects and posed significant challenges for health care providers. The maternal care provided to expectant women during the subsequent three pandemic waves is the main topic of this study.

The objective of this retrospective longitudinal study is to assess the impact of the COVID-19 pandemic on maternal care through the use of "108" ambulance services by pregnant women in normal labour and emergency conditions, as well as the antenatal care provided to pregnant women during the pandemic. Further, the study sheds light on how the policy intervention of inducting additional ambulances and recouping the medical facilities during Wave II improved emergency medical care for all, including pregnant women. The state-level 108 ambulance call centre register is used in the investigation to describe the pattern of calls for ambulance services and the utilisation of ambulance services for all medical problems, including obstetric disorders. The paper uses counterfactual analysis to assess the causal impact of the pandemic and employs a hybrid network model, a feedforward neural network in combination with an Auto Regressive (AR) Net to predict the daily count of calls for ambulance services. To the best of the authors' knowledge, this is the first study of its kind based on primary data to assess the effects of all three pandemic wave periods, including the post-pandemic phase, in developing countries. This is achieved through a novel analysis of understudied issues specific to eastern economies, such as those caused by large rural areas, joint family neighbourhoods, and the combined presence of modern allopathic medical treatment facility centres in urban areas and traditional ayurvedic and siddha practices.

## Materials and methods

### Study design

The research employs counterfactual analysis in an observational study based on empirical data from ambulance calls. The data on ambulance calls came from the 108 Ambulance control centre, which is part of the Tamil Nadu Health Services Project run by the Department of Health and Family Welfare of the Tamil Nadu government. The toll-free number is 108, and the service is available 24 hours a day, seven days a week. The state of Tamil Nadu, India, serves as the unit of study, with data collected from January 1, 2016 to December 31, 2022, spanning both the pre-pandemic period and different phases of three pandemic waves. As stated above, the research employs a hybrid neural network model (Feedforward in combination with an Auto-Regressive Neural Network) for counterfactual prediction.

### Data

**Call for ambulance service (CAS).**   These services are availed using "108 calls" where the phone calls requesting ambulance services are recorded. The '108' call centre-based ambulance system is India's centralised free-of-cost emergency response system. Primarily designed to attend to critically ill patients such as trauma victims, it is also a primary mode of transport for pregnant women in need of emergency and hospital care [22, 23]. People who are in immediate and acute need of medical attention for a variety of critical emergencies are the ones who make the CAS. The large proportion of the calls is classified as trauma, with the request to transport an injured person involved in a road crash to the nearest and most appropriate medical facility. Other types of emergent situations include calls regarding pregnancy, accidental poisoning, cerebrovascular accidents, cardiac and respiratory ailments, acute abdominal pain, fires and burns, and assaults.

The agent in the control room acquires the necessary details about the emergency, following which the nearest ambulances are diverted to the caller's location, with the control room establishing direct contact between the caller and the ambulance personnel. In addition to demographic data such as age and gender, multiple factors such as the call time, assigned time, and distance travelled were collected. The other data that is collected pertains to infection and fatality rate of the Covid-19 pandemic.

CAS are divided into two broad categories namely Non-IFT calls, where the person is transferred in an ambulance from the place of residence or trauma to the nearest healthcare facility, and IFT calls where the person is transferred from one health care facility to another for provision of appropriate medical care.

**Calls for ambulance services–pregnancy (Emergency).**   Preterm birth morbidity and mortality reduction interventions can be classified as primary (targeting all women), secondary (focusing on eliminating or reducing existing risk), or tertiary (intended to improve outcomes for preterm infants). The study, through the data provided by the CAS related to pregnant women can be classified on the basis of the above interventions. The calls related to all pregnant women requiring urgent and immediate care for delivery and other related complications, which is the primary intervention is denoted as Pregnancy (Emergency). The secondary and tertiary care interventions are covered under the calls made by pregnant women who are the beneficiaries of the antenatal program of the government. They have been screened and identified on the basis of risk and potential for complications. For the purpose of the study, it is referred as Planned ANC calls for ambulance service.

**Calls for ambulance service- pregnancy (Planned)-antenatal care.**   The "planned pregnancy calls" are requests for ambulance services to provide antenatal care to a mother during her pregnancy, thereby promoting the health of the woman and her unborn child. It is important to note that these calls were not emergency in nature but rather scheduled four ante-natal check-ups for pregnant women.

**Calls for ambulance services- vehicular trauma (Emergency).**   Road crashes are classified into three types namely fatal, serious injury, and minor injury. The victim or any bystander observing the crash dials 108 for ambulance service, and the injured is transferred to the nearest medical facility. These calls include both IFT and Non-IFT.

**Timeline of study.**   In the first week of March in 2020, the COVID-19 outbreak first appeared in the state of Tamil Nadu approximately 6–8 weeks later than in most European countries and was initially confined to the city of Chennai. This lead time presented an opportunity for the Indian government to increase its resources and readiness in the effort to contain the spread of the infection, enforce a timely and tested stay-at-home order, and additionally gear up to ensure the provision, supply, and service of emergency commodities, including medicine. The subsequent pandemic wave in 2021 was a more catastrophic one, and the third wave, which occurred in 2022, was relatively mild. The timelines for the pre-pandemic phase and the multiple phases of the COVID-19 pandemic are represented in Fig 1. Though timeline of the study extends from March 23, 2020 to December 31, 2022, the research relies on the historical data from January 1, 2016, to build the forecasting model utilizing the univariate time series data from January 1, 2016 to December 31, 2019 as training data. In Fig 2, it is seen that the first wave period lasted from March 23, 2020, to September 30, 2020, based on the infection and fatality rates, and the post-wave-I period started on October 1, 2020, and lasted until March 31, 2021. The second wave was more infectious and virulent both in terms of rate geographic and rate of spread. The second wave's time windows and its follow-up period are, respectively, from April 1 to September 30 and from October 1 to December 20, 2021. Between December 21, 2021, and March 11, 2022, wave-III had a minor impact. The time period from March 12, 2022, to September 30, 2022, may be regarded as the post-pandemic period.

## Methods

**Model training.**   To train the data, the paper employs a hybrid deep neural network model consisting of a feed-forward network and an AR-net [24]. For time series analysis, the authors employed the Neural-Prophet (N-Prophet) algorithm [25] by Meta. This model, like other

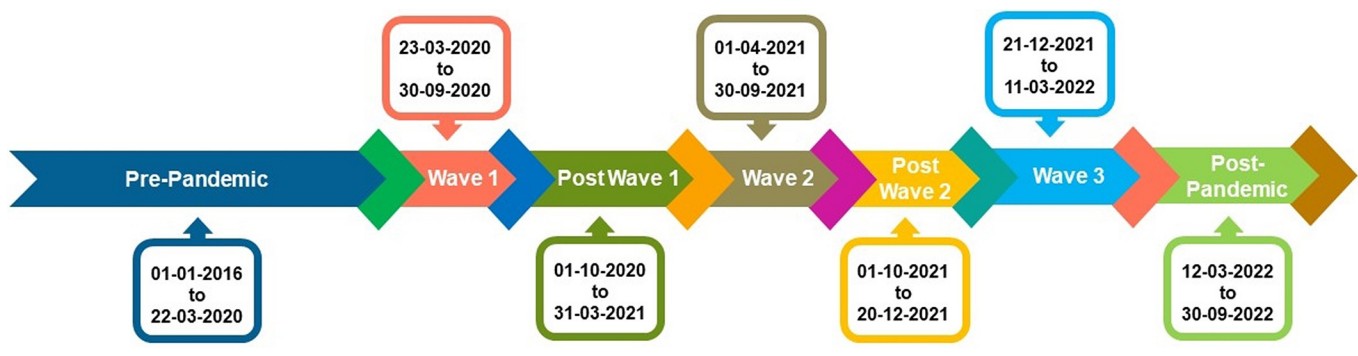

**Fig 1. Timeline of various pandemic waves in Tamil Nadu, India.**

deep learning models, automatically learns temporal dependence. The N-Prophet model is particularly useful when addressing India's seasonality problem and the moving holiday effect. Using this model in numerous real-world domains, such as traffic congestion data, power load

**Fig 2. The daily count of infection and fatalities due to COVID -19 during the three pandemic waves in Tamil Nadu.**

prediction, and even finance, yielded promising results [26, 27]. WMAPE (Weighted Mean Absolute Percentage Error) is the metric that was chosen to assess the performance of the model.

$$WMAPE = \frac{\Sigma(|Actual - Predicted|)}{\Sigma|Actual|} \tag{1}$$

The forecasted values are calculated by the following formula, considering six main factors.

$$\hat{y}_{(t+k-1)} = T_{(t+k-1)} + S_{(t+k-1)} + E_{(t+k-1)} + F_{(t+k-1)} + A_{(t+k-1)} + L_{(t+k-1)} \tag{2}$$

*Trend* $\{T_{(t+k-1)}\}$. The first factor is the trend of the temporal structure. The model determines where there are clear variations in trend and labels these as change points. The trend between these points is linear (either increasing or decreasing). This is analogous to dividing the data into multiple logistic regressions, with the cost of each logistic regression being added to the total loss. Gradient descent minimises this loss, consequently enhancing our regression. In addition, L1 regularisation (Lasso Regression) is performed to eliminate unnecessary change points.

*Seasonality* $\{S_{(t+k-1)}\}$. Seasonality denotes the periodic changes in the time series caused by its inherent characteristics, such as an increase in temperature during the summer seasons. The seasonality can be daily, monthly or even yearly. To combat seasonality, N-Prophet assumes the time series as a periodic and continuous function, which can be represented as a **FOURIER SERIES**.

$$f_p(t) = \frac{a_0}{2} + \sum_{j=1}^{\infty} \left[ a_j - \cos\left[\frac{2\pi_j t}{p}\right] + b_j \sin\left[\frac{2\pi_j t}{p}\right] \right] \tag{3}$$

Seasonality can be detected at various time periods and the value of (p) varies accordingly. For daily(p = 1), weekly(p = 7), monthly(p = 30) and yearly(p = 365), where the time step is of one day.

*Auto-regression* $\{A_{(t+k-1)}\}$. Auto regression is the process of defining future values based on past linear values, which demonstrate the dependence of time on variables part of a time series. The number of past values considered determines the model's order which is referred to as AR (p). Traditionally, AR models predict one-time step at a time, and if we need to predict *h* time steps in the future, we must fit *h* different models. The Auto-Regressive model in NeuralProphet is based on a modified version of AR-Net [24]. The classic Auto Regression process can be described using the equation,

$$y_t = c + \sum_{i=1}^{i=p} \theta_i \cdot y_{t-i} + e_t \tag{4}$$

Where,
$c$ = Intercept,
$e_t$ = White noise term

*Future regressors* $\{F_{(t+k-1)}\}$. To fit future regressors, the past and future values of that regressor must be known. These future regressors by default have an additive effect, but it can be changed to have a multiplicative effect as well. The effect of all the future regressors at a time step *t* can be denoted by the following equation,

$$F(t) = \sum_{f \in R} F_f(t) \tag{5}$$

Where,
$t$ = Time Step,

$$F_f(t) = d_f f(t)$$

$d_f$ = Coefficient of the model for future regressors

*Lagged regressors* {$L_{(t+k-1)}$}. Lagged regressors are regressors that are used to tally all of the observable variables to the time series in question, also known as covariates. The future of these regressors is unknown to us at the time of forecasting; at *t (the time* of forecasting), we have knowledge of only the previous values of the observable variables, up to the time step *t-1*.

*Events and holidays* {$E_{(t+k-1)}$}. Holiday effects, religious festivals, or any other unusual occurrence could occur inconsistently and create a bias in the forecast. Neural-Prophet allows the model to integrate two types of events: (i) user-defined events, where we can provide the model with particular information about an unusual occurrence, or (ii) country-specific holidays, where the model automatically considers the national holidays when provided the name of a country. Similar to seasonality, this can also be classified as a multiplicative effect when the need arises and as an additive effect by default.

Neural-Prophet is an extension of the already popular Prophet model developed by Taylor and Letham at Meta (earlier Facebook), which is used to forecast data at scale by making the best use of human and automated tasks [28]. It combines the Prophet's potential to handle standard features of business time series data with AR-Net's ability to scale to long-range dependencies and high interpretability as depicted in Fig 3.

Even though "Prophet" (Generalised Additive Method) is an excellent model for forecasting time series, its limitations around critical features, such as lack of local context and extensibility present a challenge for users. Neural networks provided the perfect opportunity to overcome scalability challenges but were limited by the difficulty of model interpretability. AR-Net, the latest breakthrough in modelling non-linear time series data, bridges the gap between traditional time series methods and deep learning methods, which take into account the non-linearity present in the data that traditional methods failed to capture. It also uses simple feed-forward neural networks, promoting explanation that parallel classic time series models with added scalability benefits.

Neural Prophet combines the time series components introduced by the Prophet model with neural network modules, enabling it to fit non-linear dynamics. Since being built on PyTorch, Neural Prophet can be updated with the latest innovations in deep learning, which was a limiting factor for the Prophet model built on Stan, a probabilistic programming language. Overall, Neural Prophet abstracts a significant amount of forecasting domain knowledge while incorporating best ML practices.

*Experiments.* The CAS which are being considered in this study span from January 2016 until September 2022, with a daily frequency. The different time series of interest are the Non-IFT and IFT variants of total CAS, pregnancy-related CAS, and vehicular trauma related CAS. Authors trained separate Neural-Prophet models for each of these different time series. The training data spans the period of January 1, 2016 through December 31, 2019, while the validation data spans the period of January 1, 2020 through March 22, 2020. The remaining data from March 23, 2020 through September 31, 2022 is used to compare the predictions with the actual and understand the impact of COVID on the different variants of CAS. WMAPE is used as the performance metric to measure how well the model works. The level of error is dependent on the specific industry or application in question. In the present research, a number below 10% is regarded as highly favourable, while an error range between 10% and 20% is deemed favourable. The range of 20% to 50% might be considered a level of performance that is viewed as satisfactory, whereas any percentage beyond 50% is regarded as unacceptable.

Several hyperparameters were tuned in order to achieve optimal results, by minimising the WMAPE results for the validation period. The values of the different hyperparameters are:

Batch Size: 12

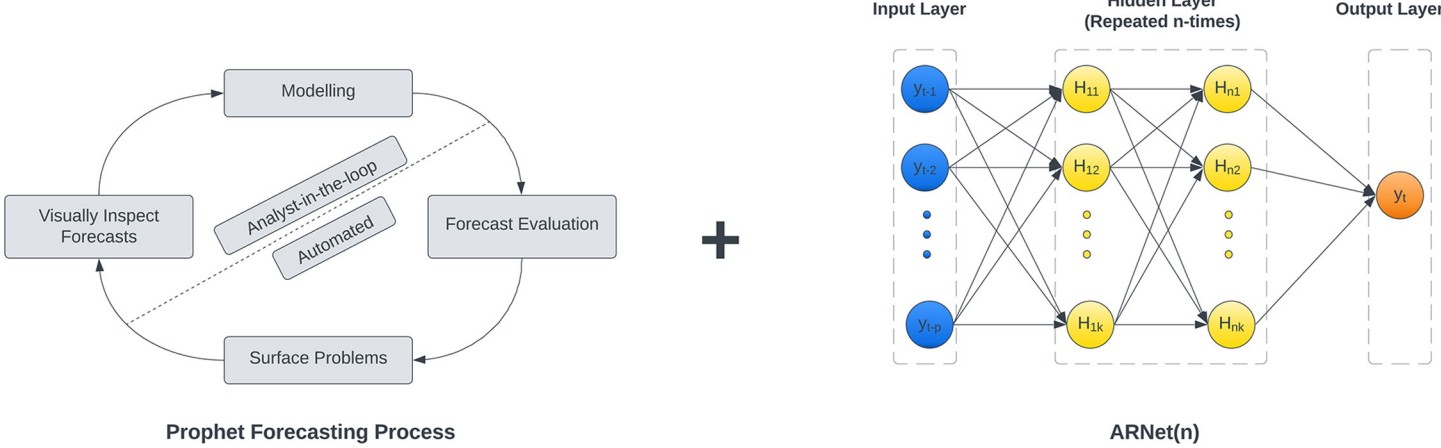

**Fig 3. The schematic representation of the architecture of the hybrid model of feedforward neural network with AR-Net.**

Learning rate: 0.01
Number of hidden layers: 10
Epochs: 40
Growth Type: Linear

Other hyper parameters such as the number of changepoints, changepoint range, number of lags, type of seasonality, seasonality mode, and trend regression were all kept at the default values and were not part of the hyperparameter tuning process.

**Effect size computation.** The authors estimate the impact of the pandemic during the three pandemic waves by comparing the actual and predicted CAS values. When the two distributions are normally distributed, the effect size is typically determined using Cohen's d. The Shapiro test was used to determine whether the actual and predicted values were normal. The Wilcoxon signed-rank test was used in place of a t-test to determine whether the observed differences between actual and predicted values were statistically significant in the event that the normality assumption was violated. The effect was calculated using Cliff's Delta, a non-parametric technique. Cliff's Delta does not make any assumptions about the distribution of the observations because it is non-parametric. The actual and predicted values are contrasted to derive the estimate for Cliff's Delta. Eq (6) provides the formula to calculate the statistic, where $x_i$ and $x_j$ are the scores within the two groups, actual and predicted, while m and n are the respective sizes for the two groups. In this study, both m and n are the same. The cardinality or count is represented by the symbol #.

$$\delta = \frac{\#\left(x_i > x_j\right) - \#\left(x_i < x_j\right)}{mn} \tag{6}$$

According to Hess and Kromrey (2004) [29],the effect size has been categorized into 3 categories for better understanding as follows:

Negligible: $|\delta| < 0.147$
Small: $0.147 = < |\delta| < 0.33$
Medium: $0.33 = < |\delta| < 0.474$
Large: $0.474 = < |\delta|$

## Results

### Accuracy of prediction model in counterfactual analysis

Additionally, to provide context, this section begins with the results of a descriptive statistical analysis and then moves on to a counterfactual. The counterfactual analysis's prediction accuracy, which was performed using a hybrid feedforward neural network and AR-Net model, was reasonably good in the range between 10% and 15%. Please see Table 1 for information on the accuracy of different univariate time series of calls. The model's accuracy was measured using the error metric known as the WMAPE during the validation period from January 1 to March 22, 2020, when the model was tested on unobserved data. In general, if the prediction length is too long, even the most sophisticated model experiences the drawback of lower accuracy. Consequently, the study's prediction length ends on September 30, 2022 (Post-Wave III phase).

### Total calls for ambulance services involving all kinds of emergencies

**Descriptive analysis-total ambulance calls for all emergencies.**  Examining the various types of emergency calls made by residents of Tamil Nadu revealed the following: Pregnant women, victims of road traffic accidents, patients with fever or infection, and those with severe abdominal pain make up the majority of those who seek the assistance of ambulance services during normal and pandemic phases. However, during the pandemic requests for COVID-19 ailments outnumbered other calls. From the pre-pandemic period to the various phases of Wave I, the total daily count of CAS has been rising gradually. However, wave II has seen a steep upward trend with a 42% increase from the daily count of 3727 calls to 5291 calls. The component of non-IFT calls has always been uniformly higher than the IFT calls in all the periods, in a range of ratios from 1.46 to 1.78. (Table 2)

Initially, the resultant strict lockdown reduced the mobility of people and vehicles to a minimum. Consequently, trauma vehicle calls decreased significantly as most roads were devoid of vehicles other than those involved in pandemic-related work. In the periods that followed the first wave, the majority of emergency calls exhibited an upward trend. During the second post-wave, the incline of these calls was steeper. As expected, there was a decline after the third wave (Fig 4).

Pregnant women who visit a clinic in person to receive diagnostic procedures and treatments are primarily covered by the CAS for ANC. These calls are not emergencies, but the ANC program's identified beneficiaries did use the 108 ambulance services. These daily CAS were typically between 2800 and 3000, but after the addition of more ambulances to the fleet in the last quarter of 2020, the number of calls rose to around 4000. During wave-II, the number of calls to ANC increased significantly by 47% compared to pre-pandemic levels. The elevated

**Table 1.  Weighted Mean Absolute Percentage Error (WMAPE) in respect of time series data for various types of calls for ambulance services in the validation phase.**

| Sl. No | Type of Call for Ambulance Service (CAS) | Validation WMAPE (Error) |
|:------:|------------------------------------------|:------------------------:|
| 1 | Total Calls | 0.069 |
| 2 | Total IFT Calls | 0.109 |
| 3 | Total NIFT Calls | 0.106 |
| 4 | Total IFT Pregnancy Calls | 0.156 |
| 5 | Total NIFT Pregnancy Calls | 0.144 |
| 6 | Total IFT Vehicular Calls | 0.145 |
| 7 | Total NIFT Vehicular Calls | 0.144 |

**Table 2. Mean daily count of total calls for ambulance services during three pandemic waves.**

| Emergency Types | Period | Total | | Inter-Facility Transfer (IFT) | | Non- Interfacility Transfer (Non-IFT) | |
|---|---|---|---|---|---|---|---|
| | | Mean | Std. dev. | Mean | Std. dev. | Mean | Std. dev. |
| **All** | Pre-Pandemic | 3338.34 | 554.19 | 1356.03 | 205.92 | 1982.31 | 449.97 |
| | Wave 1 | 3554.97 | 491.9 | 1389.64 | 185.46 | 2165.34 | 372.5 |
| | Post Wave 1 | 3727.53 | 698.92 | 1512.85 | 175.41 | 2214.68 | 604.54 |
| | Wave 2 | 5291.95 | 1299.32 | 2147.59 | 414.16 | 3144.36 | 1238.64 |
| | Post Wave 2 | 5679.21 | 1528.35 | 2036.64 | 143.51 | 3642.57 | 1475.89 |
| | Wave 3 | 5188.74 | 1266.16 | 1873.61 | 154.64 | 3315.14 | 1192.27 |
| | Post-Pandemic | 5264.2 | 2008.49 | 2078.7 | 542.83 | 3185.49 | 1659.42 |
| **Pregnancy related** | Pre-Pandemic | 789.26 | 370.13 | 438.51 | 80.48 | 352.41 | 326.68 |
| | Wave 1 | 747.88 | 138.83 | 507.71 | 77.84 | 255.2 | 89.16 |
| | Post Wave 1 | 718.51 | 128.79 | 506.19 | 95.89 | 230.88 | 54.88 |
| | Wave 2 | 814.63 | 227.75 | 531.93 | 108.96 | 301.14 | 156.24 |
| | Post Wave 2 | 1042.15 | 186.34 | 650.53 | 85.33 | 413.21 | 145.41 |
| | Wave 3 | 827.52 | 143.58 | 516.53 | 71.45 | 328.99 | 104.5 |
| | Post-Pandemic | 1002.56 | 329.61 | 650.52 | 197.55 | 373.41 | 175.65 |
| **Trauma (Vehicular)** | Pre-Pandemic | 619.48 | 142.02 | 135.69 | 35.12 | 483.79 | 114.22 |
| | Wave 1 | 283.52 | 134.1 | 65.9 | 30.08 | 217.62 | 105.81 |
| | Post Wave 1 | 694.69 | 177.36 | 155.27 | 47.12 | 539.41 | 136.3 |
| | Wave 2 | 574.81 | 242.2 | 141.4 | 63.69 | 433.41 | 182.2 |
| | Post Wave 2 | 725.56 | 202.17 | 179.86 | 48.57 | 545.69 | 155.72 |
| | Wave 3 | 788.16 | 184.78 | 193.9 | 45.78 | 594.26 | 142.07 |
| | Post-Pandemic | 875.18 | 271.19 | 221.47 | 75.62 | 663.67 | 205.32 |
| **Acute Abdomen** | Pre-Pandemic | 288.28 | 48.35 | 113.45 | 25.85 | 174.83 | 31.18 |
| | Wave 1 | 140.3 | 55.39 | 43.93 | 11.53 | 96.37 | 47.24 |
| | Post Wave 1 | 220.15 | 72.82 | 88.98 | 34.15 | 131.16 | 42.21 |
| | Wave 2 | 223.82 | 78.61 | 76.61 | 30.34 | 147.21 | 51.54 |
| | Post Wave 2 | 340.98 | 40.79 | 124.46 | 23.21 | 216.52 | 26.94 |
| | Wave 3 | 331.02 | 33.25 | 124.41 | 17.6 | 206.62 | 26.55 |
| | Post-Pandemic | 365.17 | 94.23 | 163.35 | 49.81 | 208.2 | 52.47 |
| **COVID-19** | Pre-Pandemic | 0 | 0 | 0 | 0 | 0 | 0 |
| | Wave 1 | 1258.6 | 701.98 | 346.17 | 198.34 | 919.16 | 528.79 |
| | Post Wave 1 | 279.26 | 280.72 | 130.37 | 116.03 | 148.9 | 167.3 |
| | Wave 2 | 1394.85 | 1195.97 | 771.42 | 649.05 | 623.43 | 560.52 |
| | Post Wave 2 | 197.81 | 134.78 | 105.15 | 42.2 | 92.67 | 114.55 |
| | Wave 3 | 260.65 | 251.34 | 144.89 | 141.93 | 117.21 | 112.92 |
| | Post-Pandemic | 12.59 | 14.03 | 9.81 | 10.22 | 5.44 | 6.12 |

levels of calls relative to pre-pandemic periods were sustained during the post-wave-II, wave-III, and post-wave-III periods with increases of 51%, 38%, and 38%, respectively.

After the state experienced a wave of COVID-19 cases during the first pandemic wave, the fleet strength of the 108 ambulance services, which had been around 900 during the previous few years, was significantly increased to 1300. In the month of April 2018, a total of 931 ambulances were available. Subsequently, in April 2020, the first addition of an ambulance during the pandemic occurred, resulting in a fleet strength of 991. During the subsequent months, a limited number of additional ambulances were incorporated into the fleet. The most significant augmentation occurred in November 2020, resulting in a cumulative total of 1300

## Comparison of Mean of Calls for Emergency Types in Total Calls

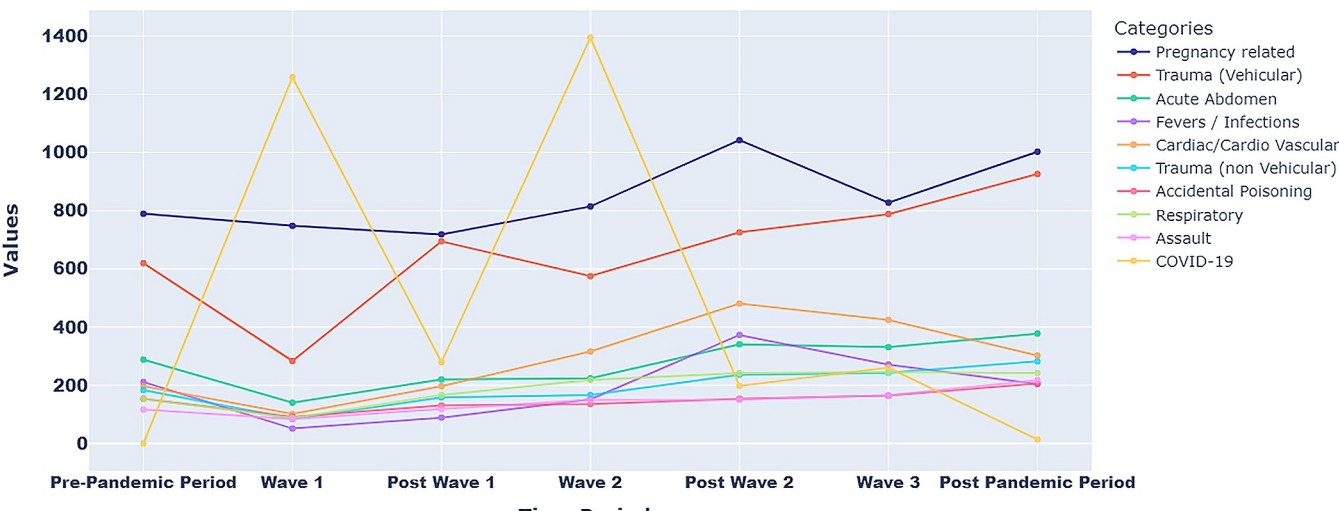

**Fig 4. Various categories of total emergency calls during three pandemic waves.**

ambulances. During the period from July 2021 to February 2022, there was a slight decrease in the fleet size, with the number of ambulances approximately amounting to 1230. In March 2022, a significant increase was observed, resulting in a total fleet strength of 1353. The average distance an ambulance travelled on a trip during the pre-pandemic and wave-I phases was 8.81 kilometres (km) and 11.64 km, respectively. Because of increased fleet strength, this distance was reduced significantly in post-wave II to 6.99 km, thanks to the additional fleet numbers. Similarly, the time taken for reaching the victims also decreased significantly, from 21.9 minutes in the pandemic era to 12.93 minutes in wave III. (Refer to Table 3). The average distance travelled includes three distinct ambulance moves. First, the ambulance travels from its base location to the spot of the caller. The second factor is the distance between the caller's location and the nearest hospital. The ambulance finally returns to its original base location after leaving the hospital.

Prior to the pandemic, the average age of female callers was 32.17 years; however, during the pandemic phases, particularly during wave-II, the average age increased to 35.98 years as more victims requested COVID-19-related assistance than they had in the earlier phase, when a significant proportion of callers were pregnant women. Similar to this, the average age of

**Table 3. Mean, median and mode of the distance and time taken for total calls for ambulance services and number of antenatal care service calls during the three pandemic waves in Tamil Nadu.**

| Period | Distance Covered (Km) | | | Time taken to reach scene (Minutes) | | | Number of ANC Calls | | |
|---|---|---|---|---|---|---|---|---|---|
| | Mean | Median | Mode | Mean | Median | Mode | Mean | Median | Mode |
| Pre-Pandemic Period | 8.81 | 8.81 | 6.32 | 21.9 | 11.72 | 2 | 2939.17 | 2926 | 2588 |
| Wave 1 | 11.64 | 11.36 | 8.72 | 18.43 | 12.57 | 1.98 | 2967.89 | 2909.5 | 2292 |
| Post Wave 1 | 8.12 | 8 | 5.95 | 13.4 | 6.85 | 1.98 | 3026.59 | 3001.5 | 2751 |
| Wave 2 | 8.24 | 7.7 | 4.93 | 17.88 | 6.72 | 1.98 | 4336.39 | 4302.5 | 3966 |
| Post Wave 2 | 6.99 | 7.2 | 5.12 | 13.34 | 6.28 | 1.98 | 4450.64 | 4378 | 4285 |
| Wave 3 | 7.1 | 7.14 | 5.29 | 12.93 | 6.78 | 1.98 | 4046.82 | 3960 | 3670 |
| Post-Pandemic Period | 7.56 | 7.68 | 5.72 | 13.66 | 8.33 | 2 | 3941.6 | 3620.5 | 3502 |

male callers rose from 40.33 years during the pre-pandemic period to 43.55 years during COVID-19 wave-II. (See Table 4).

## Counterfactual analysis- total ambulance calls for all emergencies

The daily count of total CAS for all kinds of emergencies could be used as a proxy variable to gauge the workload pressures on the hospital's infrastructure. The counterfactual analysis predicted 2182 to 2543 CAS per day from the scene of trauma or person's home for Non-IFT, whereas the recorded CAS during the same period was 2165 to 3642. The counterfactual analysis shows that the first wave of COVID-19 had no effect on daily CAS, as there was only a 1% drop in actual CAS compared to the counterfactuals and a reported very low Cliff's delta of -0.086 that measures the difference between the actual and predicted distributions. In the phases that followed wave-I, the actual reported CAS indicated an upsurge with reference to the counterfactual predicted calls from the post-wave-I phase to wave-III. The percentage increase was between 36% and 43%, and the increase in effect size (Cliff's delta) was between 0.652 and 0.77 (Fig 5).

Callers planning to look for an inter-facility transfer, usually from one hospital to another with the right medical and treatment capabilities or for other reasons like personal choice, proximity to their home, etc., followed a similar pattern. Even though they were not very different from non-IFT calls, the effects of IFT calls were more noticeable in each phase of the pandemic waves. During wave I, the actual calls were 10% higher than the counterfactual calls, which was measured as a Cliff's Delta of 0.452. In the periods after wave I, the percentage increase in actual calls compared to the counterfactual ranged from 34% to 142%, and the effect size ranged from 0.954 to 1.0. (Table 5 and Fig 5)

## Calls for ambulance services–pregnancy (Emergency)

**Descriptive analysis- ambulance calls for pregnancy (Emergency).**   Fig 6.

Pregnancy-related emergency calls followed a pattern that was quite reminiscent of non-vehicle trauma, accidental poisoning, and respiratory illnesses. The number of CAS decreased from 789 to 747 to 718 between the pre-pandemic, wave-I, and post-wave-I periods. This period of wave-I was exceptional, as the entire state came to a standstill when the government imposed a strict lockdown and everyone, with the exception of emergency services, remained indoors. During wave-II, post-wave-II, wave-III, and post-wave-III, however, pregnancy-related calls increased. Compared to the post-wave-I (718 calls), the increase was notable and significant during the post-wave-II (1042 calls) and post-wave-III (1002 calls) periods.

In contrast to other types of emergencies, the proportion of IFT pregnancy-related calls exceeded that of non-IFT calls. Non-IFT and IFT calls reflect a similar overall trend of pregnancy-related calls, with the exception of wave-I, in which Non-IFT calls decreased while IFT calls increased marginally. In addition, the increase in Non-IFT calls during the pre-pandemic, wave-II, and wave-III periods was less than the increase in IFT calls during the same time period. Non-IFT calls rose from 352 during the pre-pandemic period to 373 following wave-III. During the same time frame, the number of IFT calls increased from 438 to 650.

**Counterfactual analysis - ambulance calls for pregnancy (Emergency).**   The findings of the investigation based on counterfactual predictions are nearly identical to the trends observed in the descriptive statistical analysis. In the descriptive study, the comparison was based on the pre-pandemic period. However, here in the present analysis, the results indicate a decrease or increase in calls relative to the counterfactual, i.e., had there been no pandemic or no change in the setup of the ambulance services. In the Non-IFT category of calls, there was a medium effect size decrease (Cliff's Delta of -0.335, -0.466) and negligible and medium

**Table 4. Mean, median and mode of the age of the callers for total calls for ambulance services for different genders.**

| Period | Total Victim Age (Years) | | | Male Victim Age (Years) | | | Female Victim Age (Years) | | | Transgender Victim Age (Years) | | |
|---|---|---|---|---|---|---|---|---|---|---|---|---|
| | Mean | Median | Mode | Mean | Median | Mode | Mean | Median | Mode | Mean | Median | Mode |
| **Pre-Pandemic Period** | 36.05 | 36.21 | 30.23 | 40.32 | 40.38 | 40.46 | 32.17 | 32.3 | 26.33 | 34.74 | 29 | 25 |
| **Wave 1** | 36.78 | 36.8 | 33.43 | 41.11 | 41.02 | 38.22 | 32.89 | 32.87 | 29.25 | 31.13 | 29.46 | 20 |
| **Post Wave 1** | 37.4 | 37.9 | 31.27 | 42.28 | 42.4 | 37.69 | 33.06 | 33.41 | 27.24 | 25.84 | 25 | 21 |
| **Wave 2** | 39.49 | 39.39 | 30.04 | 43.55 | 42.98 | 39.82 | 35.98 | 35.93 | 27.04 | 50.83 | 34.5 | 22 |
| **Post Wave 2** | 37.89 | 38.87 | 29.95 | 42.87 | 42.86 | 38.46 | 34.02 | 34.85 | 26.92 | 37.48 | 33 | 24 |
| **Wave 3** | 38.81 | 39.49 | 30.9 | 43.29 | 43.34 | 40.35 | 34.72 | 35.41 | 27.2 | 32.67 | 30.5 | 26 |
| **Post-Pandemic Period** | 37.43 | 38.3 | 30.08 | 42.46 | 42.44 | 37.71 | 33.66 | 34.1 | 26.9 | 39.1 | 33 | 24 |

increases (0.078 and 0.333) during wave-I, post-wave-I, and wave-II, post-wave-II periods, respectively. In the remaining phases, namely wave III and post-wave III, the escalation was reported by the Cliff's Deltas at 0.523 and 0.517, respectively. The impact of the pandemic and concomitant interventions that followed the first wave was greater for IFT calls than for Non-IFT calls. In the wave I, post-wave I, and wave II periods, there was a very large effect size increase (Cliff's Deltas of 0.769, 0.951, and 0.977, respectively) in the calls when compared with the counterfactual. In the phases that followed the second wave, extremely large +1 effect sizes were observed. The difference in mean Non-IFT calls from wave-I to post-wave-III ranged from 13% to 44%, whereas the range for IFT calls was 24% to 165% in the corresponding phases. (Table 6 and Fig 7).

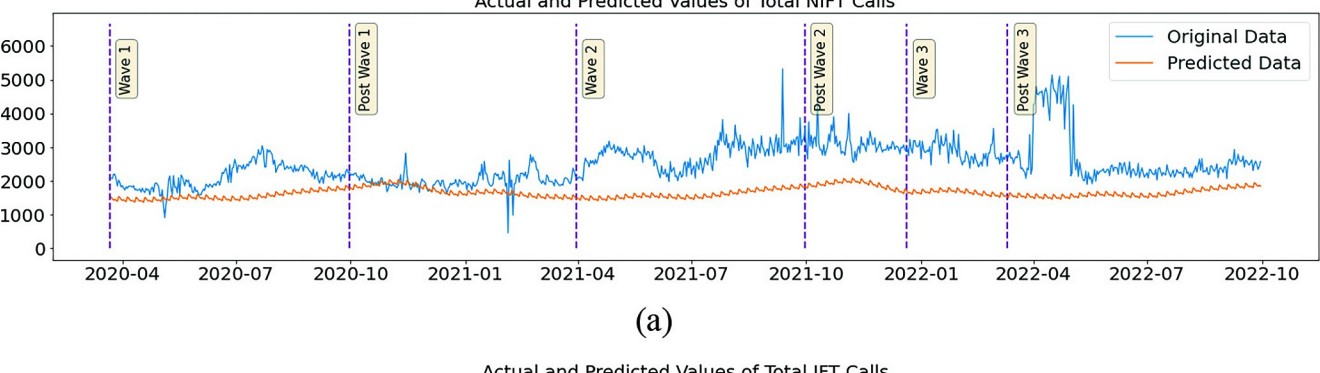

(a)

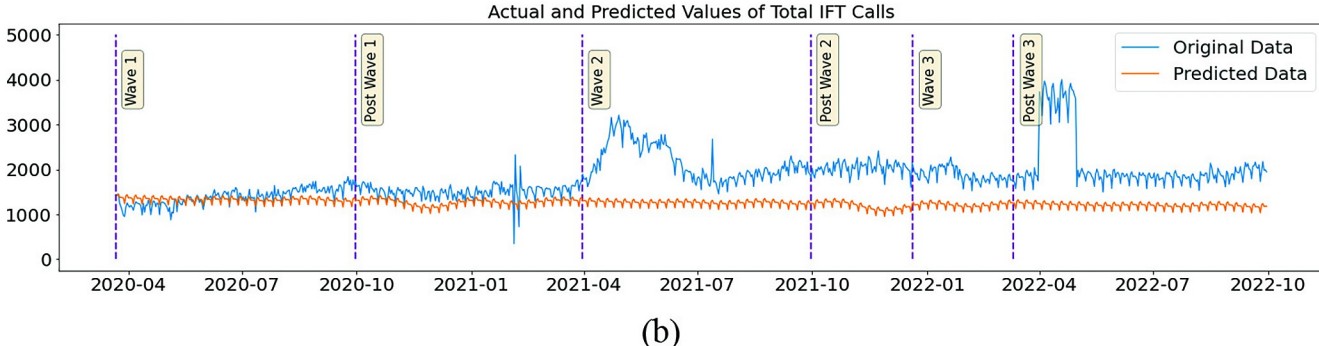

(b)

**Fig 5.** Plot of the actual and predicted daily count of calls for ambulance service (Non-IFT- Top Panel and IFT- Bottom Panel) (on Y-axis) in Tamil Nadu during the three waves of COVID-19 from 2020 to 2022 (on X-axis). The vertical bars indicate the periods of three waves. The prediction is done using a hybrid model of feedforward network with AR Net.

**Table 5. The actual and predicted daily count of the total IFT and non-IFT calls for ambulance service with percentage difference and effect size expressed in Cliff's Delta.**

| Period | Mean Actual Non-IFT | Mean Predicted Non-IFT | Percentage Diff Non-IFT | Effect Size Non-IFT | Mean Actual IFT | Mean Predicted IFT | Percentage Diff IFT | Effect Size IFT |
|---|---|---|---|---|---|---|---|---|
| Wave 1 | 2165.34 | 2182.5 | -0.79 | -0.086 negligible | 1389.64 | 1262 | 10.11 | 0.452 medium |
| Post Wave 1 | 2214.68 | 2349.516 | -5.74 | -0.718 large | 1512.85 | 1127.5 | 34.18 | 0.954 large |
| Wave 2 | 3144.36 | 2304.377 | 36.45 | 0.652 large | 2147.59 | 1066.153 | 101.43 | 1.0 large |
| Post Wave 2 | 3642.57 | 2543.407 | 43.22 | 0.77 large | 2036.64 | 996.4691 | 104.39 | 1.0 large |
| Wave 3 | 3315.14 | 2423.963 | 36.77 | 0.742 large | 1873.61 | 878.284 | 113.33 | 1.0 large |
| Post Wave 3 | 3264.06 | 2423.03 | 34.71 | 0.145 negligible | 2118.22 | 874.7734 | 142.14 | 1.0 large |

## Calls for ambulance service- pregnancy (Planned)-antenatal care

Despite being marginal, it is important to note that there was only a very slight increase in calls during wave-I, given the overall decline in calls for all types of emergencies. Expectedly, COVID-19-related illness calls are the only type of emergency that has increased during this relevant period. From pre-pandemic to wave-I, the average weekly calls increased from 2939 to 2967, and then again to 3026 in the post-wave-I period. The steep ascent of calls in wave-II and post-wave-II to 4336 and 4450, respectively, decreased marginally to 4046 and 3941 in wave-III and post-wave-III. As the ANC scheme was launched only in 2019, there is no sufficient historical data for making predictions using the neural network method.

## Calls for ambulance services- vehicular trauma (Emergency)

The comparative study of the vehicular trauma calls vis-à-vis total CAS and pregnancy CAS is done to interpret the differential impact of the pandemic on various kinds of emergencies and the response of the hospital and other allied infrastructure. As shown in Figs 4 and 8, there was steep decline in the vehicular trauma calls during the first wave, whereas such a trend was not noticed in other types of emergencies, including pregnancy.

In both the IFT and Non-IFT categories, the actual decline in calls from the pre-pandemic to wave-I phases was substantial. While the mean daily count of total vehicular trauma-related calls decreased from 619 to 283 (46%) between the pre-pandemic and wave-I periods, the mean daily count of IFT calls decreased from 135 to 66 (49%) and Non-IFT calls decreased from 484 to 213 (44%).

## Mean Daily Count of Pregnancy Related Calls

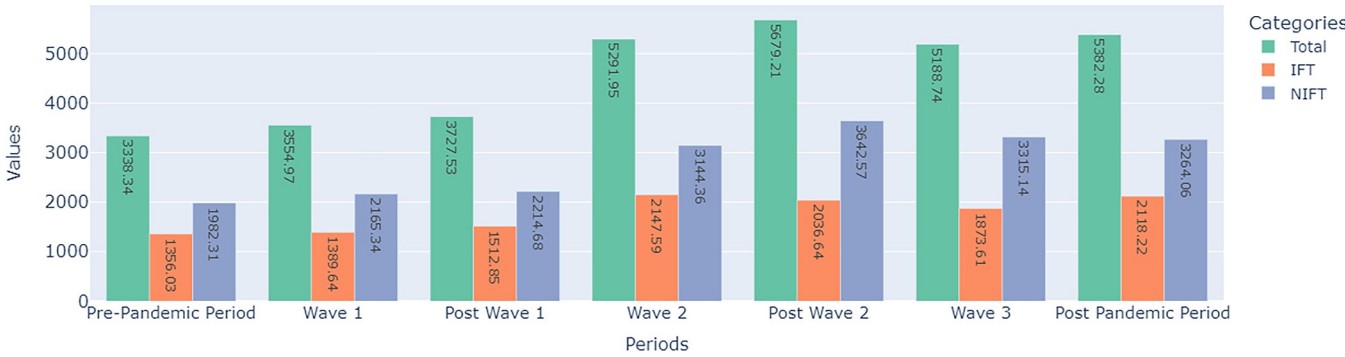

**Fig 6. Plot of total calls for pregnancy emergency related ambulance services with the break-up of non-IFT and IFT categories of calls.**

**Table 6. The actual and predicted daily count of the pregnancy related IFT and non-IFT calls for ambulance service with percentage difference and effect size expressed in Cliff's Delta.**

| Period | Mean Actual | Mean Predicted | Percentage Diff | Effect Size | Mean Actual | Mean Predicted | Percentage Diff | Effect Size |
| --- | --- | --- | --- | --- | --- | --- | --- | --- |
| | Non-IFT | Non-IFT | Non-IFT | Non-IFT | IFT | IFT | IFT | IFT |
| **Wave 1** | 255.2 | 261.76 | -2.503 | -0.335 medium | 507.71 | 407.59 | 24.56 | 0.769 large |
| **Post Wave 1** | 230.88 | 285.48 | -19.125 | -0.466 medium | 506.19 | 313.6 | 61.41 | 0.951 large |
| **Wave 2** | 301.14 | 264.55 | 13.831 | 0.078 negligible | 531.93 | 327.99 | 62.18 | 0.977 large |
| **Post Wave 2** | 413.21 | 347.06 | 19.059 | 0.333 medium | 650.53 | 249.85 | 160.37 | 1.0 large |
| **Wave 3** | 328.99 | 237.86 | 38.309 | 0.523 large | 516.53 | 214.44 | 140.87 | 1.0 large |
| **Post Wave 3** | 382.77 | 265.37 | 44.239 | 0.517 large | 661.92 | 249.63 | 165.16 | 1.0 large |

The counterfactual analysis also yields similar results for the first wave decline for Non-IFT (-44% decline and -0.884 Cliff's Delta) and IFT (-45% decline and -0.884 Cliff's Delta) CAS. The counterfactual increase during post-wave-I and wave-II periods for IFT calls was moderate, with a Cliff's Delta of approximately 0.408, whereas the corresponding increase for Non-IFT calls was significant: 0.801 during post-wave I and 0.459 during wave-II. The periods that followed, namely post-wave-II, wave-III, and post-wave-III, demonstrated a larger impact, with Cliff's Delta values ranging from 0.814 to 1.00 (a very large effect size) for both IFT and Non-IFT categories of calls. (Fig 9 and Table 7).

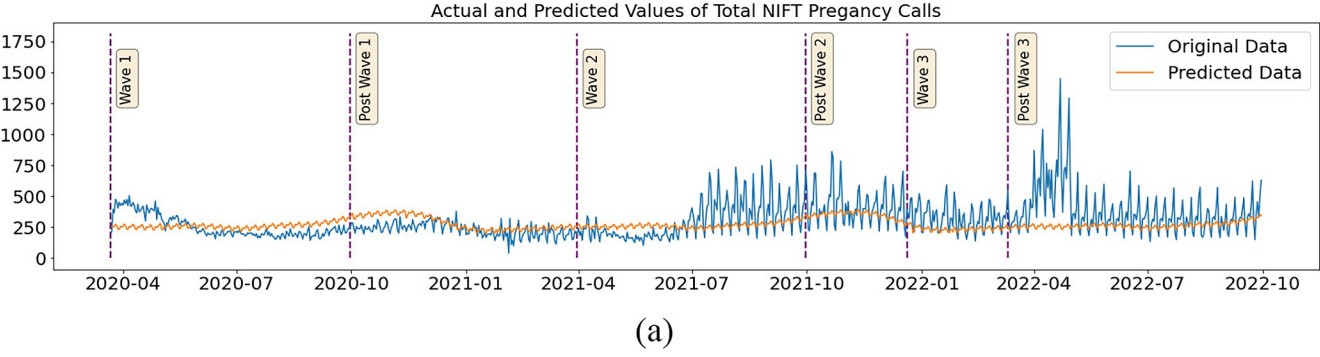

(a)

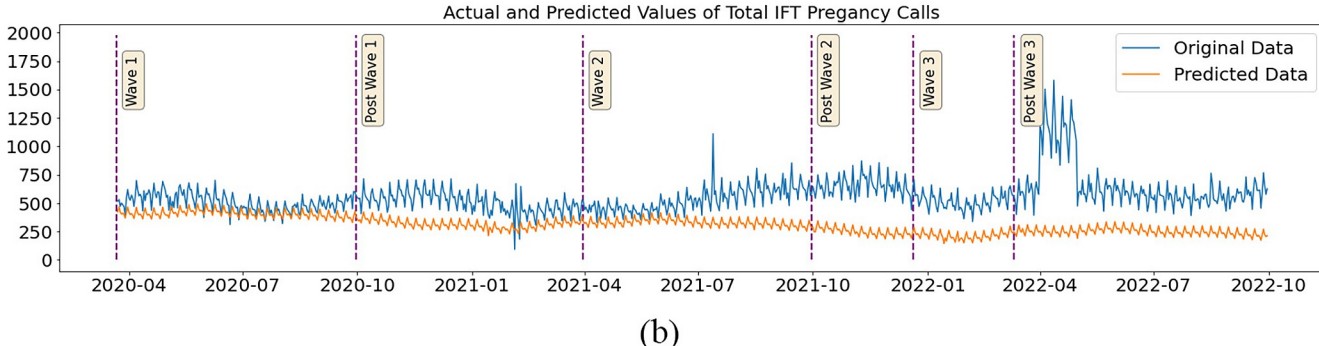

(b)

**Fig 7.** Plot of the actual and predicted daily count of pregnancy related calls for ambulance service (Non-IFT-7a and IFT-7b) (on Y-axis) in Tamil Nadu during the three waves of COVID-19 from 2020 to 2022 (on X-axis). The vertical bars indicate the periods of three waves. The prediction is done using a hybrid model of feedforward network with AR Net.

## Comparison between Total Calls for All, Pregnancy Related and Vehicular Trauma

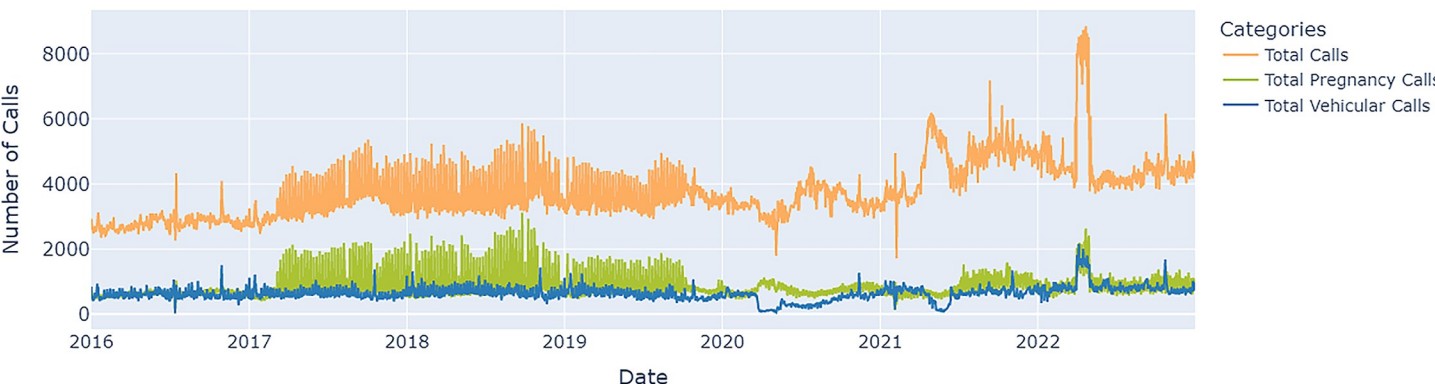

**Fig 8. Plot of the actual daily count of calls for ambulance services during the entire period covering pre-pandemic to post pandemic phases for total, pregnancy and trauma related in Tamil Nadu.**

## Discussion

The Covid 19 pandemic had a significant impact on the mortality and health care infrastructure across the globe, and governments across the world deployed various remedial and counter measures to tackle the pandemic, including lockdown measures, mass vaccination campaigns and enhancing health care infrastructures. The unknown pandemic caused

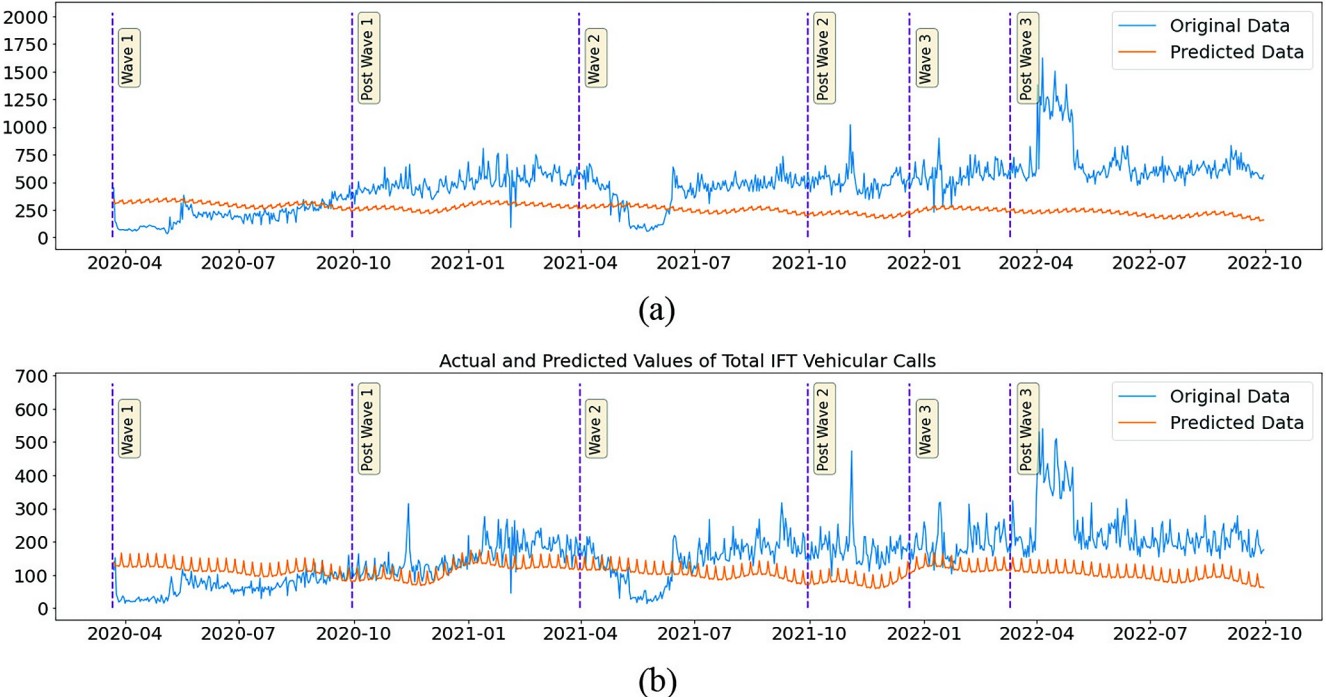

**Fig 9.** Plot of the actual and predicted daily count of vehicular trauma related calls for ambulance service (Non-IFT- Top Panel and IFT- Bottom Panel) (on Y-axis) in Tamil Nadu during the three waves of COVID-19 from 2020 to 2022 (on X-axis). The vertical bars indicate the periods of three waves. The prediction is done using a hybrid model of feedforward network with AR-Net.

**Table 7. The actual and predicted daily count of the vehicular trauma related IFT and non-IFT calls for ambulance service with percentage difference and effect size expressed in Cliffs Delta.**

| Period | Mean Actual | Mean Predicted | Percentage Diff | Effect Size | Mean Actual | Mean Predicted | Percentage Diff | Effect Size |
|---|---|---|---|---|---|---|---|---|
| | Non-IFT | Non-IFT | Non-IFT | Non-IFT | IFT | IFT | IFT | IFT |
| Wave 1 | 217.62 | 393.39 | -44.68 | -0.884 large | 65.9 | 120.15 | -45.15 | -0.884 large |
| Post Wave 1 | 539.41 | 351.7 | 53.37 | 0.801 large | 155.27 | 123.43 | 25.8 | 0.408 medium |
| Wave 2 | 433.41 | 315.96 | 37.17 | 0.459 medium | 141.4 | 112.36 | 25.84 | 0.422 medium |
| Post Wave 2 | 545.69 | 246.12 | 121.71 | 0.963 large | 179.86 | 95.43 | 88.47 | 0.967 large |
| Wave 3 | 594.26 | 304.7 | 95.03 | 0.919 large | 193.9 | 134.75 | 43.89 | 0.814 large |
| Post Wave 3 | 740.05 | 242.77 | 204.84 | 1.0 large | 236.05 | 107.23 | 120.13 | 0.997 large |

unprecedented apprehension among the populace, resulting in strict adherence to the government-enforced lockdown. Prior to the discovery of vaccination, one of the critical governmental interventions was a strict lockdown that reduced the mobility of people and vehicles to a minimum. Consequently, as Kandaswamy Paramasivan and colleagues reported, most roads were devoid of vehicles when the strict stay-at-home orders were in-force, consequently, road crashes declined phenomenally. Expectedly the trauma vehicle calls increased significantly during these periods [30]

Ram & Sornette's study designed a framework to understand and quantify the effectiveness of the interventions implemented by various countries, including China, Italy, Spain, Switzerland, France, Canada, United States, Germany, India, and Japan, to control epidemic growth [31]. They evaluated the time evolution of effective reproduction numbers in various geographical regions and then estimated the counterfactual evolution of effective reproduction number to quantify the impact of various lockdown measures. Whilst the authors observed that implementing lockdown resulted in reducing the reproduction number, this reduction was not uniform across all countries and there was a transient increase in the contagion after the lockdown. The authors also concluded that dynamics of hospital admissions would also need to be considered to draw a definitive conclusion [31]. In this context, our study assesses the impact of Covid 19 pandemic on MCH and evaluates the possible underlying factors for the derived results.

As a result of the general public's tendency to avoid hospitals and COVID-19's stringent restrictions, it was anticipated that the total number of calls for all types of emergency ambulance services would decrease. However, in view of the alarming increase in COVID-19 ailment calls, there was a moderate increase in the total number of calls for all types of emergencies when compared with the counterfactual, especially in the category of IFT CAS. This is supported by the fact that there was a negligible decrease in non-IFT CAS, as reported by the Cliff's Delta of -0.086, whereas IFT CAS moderately escalated (Cliff's Delta 0.452), as IFT calls are primarily from one facility to another, where COVID-19 restrictions and hospital-avoidance behaviour would have no effect.

Two significant differences distinguish wave-II from wave-I. The infection and mortality rates were five to six times those of the first wave. While the first wave of the pandemic mostly affected cities, the second wave, which carried a deadlier and more virulent virus, quickly spread to a wider area, including far flung corners of the state. This was exacerbated by a lack of vital medical supplies, including oxygen, and a strained hospital infrastructure. An increased number of patients needed to be hospitalized that necessitated many newer facilities to be established and some existing facilities were turned into COVID care centres exclusively [32]. The targeted vaccination of a vulnerable population had just begun in January 2021, but it was not expected to contain the disease's spread in the second wave immediately. However, in the

third wave, vaccination reduced the hospitalisation rate of infected people, thereby alleviating their suffering. The second development studied in the research is the expansion of the state's ambulance fleet, besides other measures to recoup various resources to combat the pandemic. Additionally, there was a vigorous promotional effort to encourage the use of 108 ambulances [33]. Therefore, these two factors account for the high number of actual CAS for all emergencies, with COVID-19 illness and pregnancy being the most prevalent. This is evident as the percentage actual increase in total IFT CAS when compared with the counterfactual during wave-II and subsequent periods (post-wave-II, wave-III, and post-wave-III) was greater than 100 percent, with a Cliff's-Delta effect size of one. While there was a comparable increase in total Non-IFT CAS, the magnitude was 44%, and the Cliff's Delta effect sizes were in the range of 0.65 to 0.77. Expectedly, Cliff's Delta 0.145 had no noticeable effect after the third wave. From the preceding discussion, it is evident that the hospital infrastructure was significantly overstretched at the onset of wave-II and that there was little relief for the state's overburdened medical and health apparatus until after wave-III. In addition, it is important to note that the noticeable increase in emergency calls may also be attributable to the increased fleet size of ambulances and the efforts made to promote awareness of these services to the general public in Tamil Nadu. If the severity of the pandemic were the only cause, there would not have been a sustained and continued increase in calls during wave III, which was the least dangerous of the three waves of the pandemic.

Although the trend of pregnancy emergency calls was quite similar to the overall pattern of total calls for all emergencies, their magnitude of increase was significantly greater than that of all other emergencies. Regarding IFT calls related to pregnancy emergencies, the actual calls were astronomically higher than the predicted calls; for example, in the post-wave-II, wave-III, and post-wave-III periods, respectively, they were 250%, 214%, and 250% higher. A Cliff's Delta of 1.0 was the reported effect for each of the aforementioned time periods. Although the Non-IFT category of calls showed a similar rising trend, the magnitude was much lower, ranging from 19% to 44% (Cliff's Delta 0.33 to 0.5), in the corresponding periods. In particular, when the pandemic's second wave reached its apex, there was a spike in the number of calls in both the Non-IFT and IFT categories, with the former recording a moderate increase of 14% (Cliff's Delta 0.078 negligible) and the latter registering a significant increase of 327% (Cliff's Delta 0.977). It is remarkable to note that despite the overwhelming use of hospital resources to cater to COVID-19 infected patients, the health services did respond exceedingly well in attending to pregnancy-related emergencies and also shifting patients to more appropriate facilities for better treatment. As stated earlier, this acute rise in the number of IFT calls for emergency pregnancies that began in the post-wave-II phase may be attributed to the increased accessibility of ambulance services as well as the increased popularity and awareness of the free services, especially among the women. The COVID-19 pandemic weaned off after wave-II, and it is possible that this, along with fewer and milder restrictions and a relaxation of some COVID-19-appropriate conduct and behaviour, facilitated the access to CAS for pregnant women.

During the pandemic, pregnant women were more susceptible to contracting the virus. The infection may lead to a decline in health and ultimately affect the foetus [34]. Therefore, prominence was placed on the diagnosis and management of pregnant women who may be asymptomatic or have a mild disease. Pregnant women with underlying medical conditions, such as hypertension, diabetes, and obesity, and those over the age of 35 were given special consideration. In light of the pandemic and the fact that numerous government programmes and initiatives have been shelved or postponed, one would expect that the government would not provide this antenatal care. However, it is interesting to note that antenatal care has received a substantial boost since wave-II, due to the increased fleet strength of ambulances and other

maternal care-related resources, as evidenced by a 51%, 38%, and 38% increase in calls in the post-wave-II, wave-III, and post-wave-III periods, respectively, when compared with the pre-pandemic period. The authors did not conduct a counterfactual analysis of the planned pregnancy-related care provided by the government to pregnant women who had previously been screened and identified as requiring antenatal care due to certain risks. This was due to the lack of historical data, as the government's antenatal care programme was only launched in 2019. Thus, the study specifies that emergency and prenatal care for pregnant women were not affected by the pandemic. In contrast, the number of patients treated was significantly greater, and care was sustained throughout the entire pandemic and post-pandemic period from March 22, 2020, to December 31, 2022. (Refer to Fig 10).

## Conclusions

The study used a hybrid model of a feed-forward neural network and an AR-net to predict counterfactual events using primary data from ambulance calls. This model provided excellent accuracy and interpretability, overcoming the shortcomings of the poorer accuracy of conventional models and the complexity and non-interpretability of deep learning models.

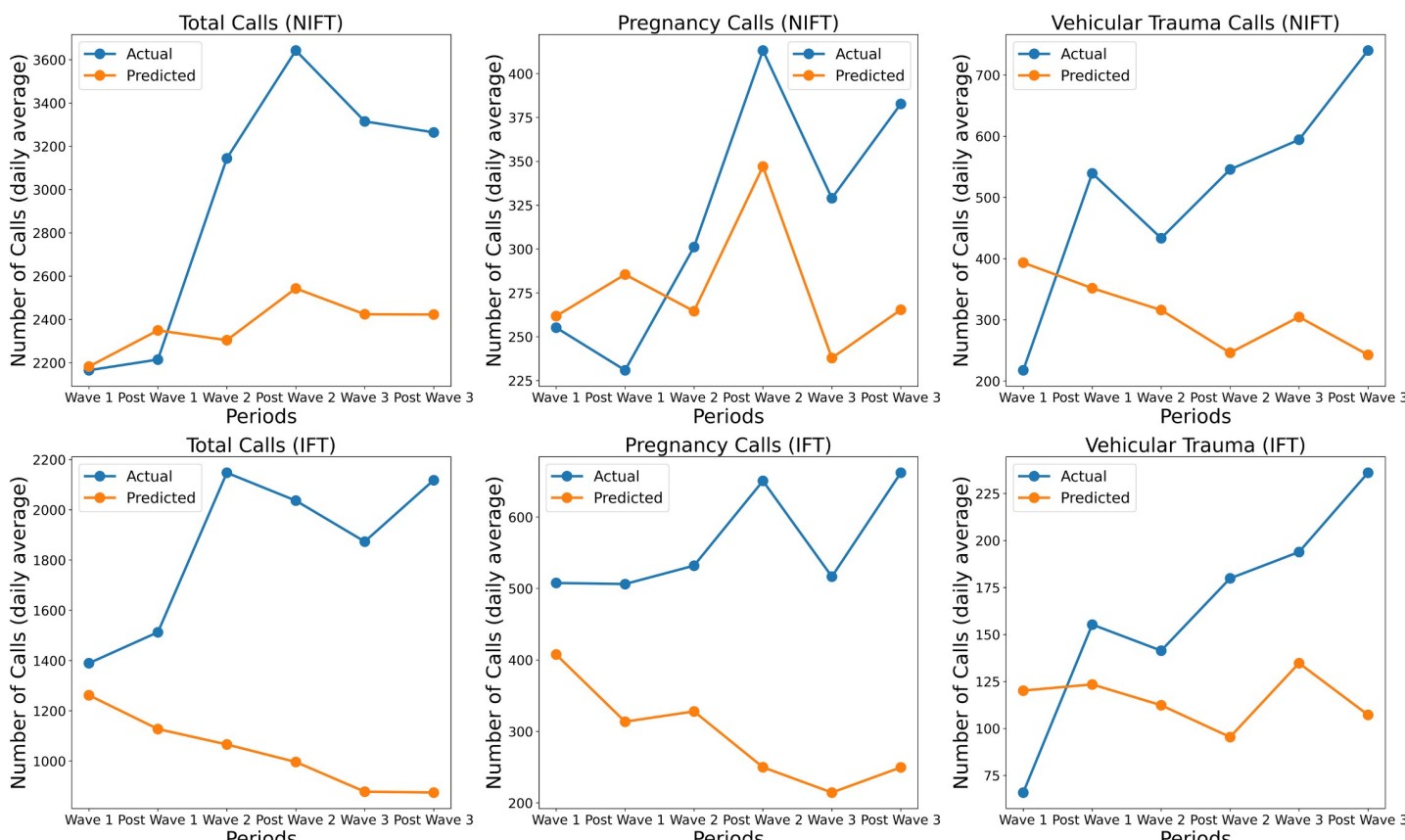

**Fig 10.** Plot (Discrete) of the actual and predicted daily count of total calls, pregnancy related calls, trauma-vehicular calls (on Y-axis from Left to Right) in Tamil Nadu during the three waves of COVID-19 from 2020 to 2022 (on X-axis). The vertical bars indicate the periods of three waves. Top Panels pertain to Non-IFT calls and Bottom Panels refer to IFT Calls. The prediction is done using a hybrid model of feedforward network with AR Net.

The second wave of the pandemic, which occurred in 2021, was the most devastating in Tamil Nadu, affecting a broader cross-section of the population as opposed to the first wave, which primarily affected the elderly and urban dwellers. When compared to the counterfactual, the phases in wave-II, post wave-II, wave-III, and post wave-III saw a significant increase in both the total IFT and total Non-IFT calls covering all emergencies, as indicated by the reported Cliff's Delta values of 1.0 and the Cliff's Delta range of 0.65 to 0.77, respectively. These findings highlight the health system's and other allied apparatus's overstretched resources during the intense Wave II and post-Wave II periods.

The pandemic had no effect on maternal care for emergencies involving pregnant women and infants, despite the dedication of health workers to pandemic work, as stated above. In contrast, a substantial improvement in performance was observed, particularly in the area of dedicated and sophisticated medical care in an appropriate medical facility, as evidenced by the IFT calls in wave-II, post-wave-II, wave-III, and post-wave-III, which saw respective increases of 62%, 160%, 140%, and 165% when compared to the counterfactual (Cliff's Delta 0.977, 1, 1, and 1).

Despite the extreme severity of the pandemic in wave-II, when the health services were fully committed to COVID-19 duties, it is worth noting that antenatal care has received a significant boost since wave-II, owing to increased fleet strength of ambulances and other maternal care-related resources, as evidenced by a 47%, 51%, 38%, and 38% increase in calls in the wave-II, post-wave-II, wave-III, and post-wave-III periods relative to the pre-pandemic period. The expansion of ambulance services and promotion of the 108 ambulance services scheme during the wave-II and post-periods resulted in improved emergency care for all, including mothers. Authors believe that the research findings provided here would help public health and government authorities in the planning and delivery of vital maternal and child health care that could be resilient even in the times of pandemic.

## Supporting information

**S1 File. Background- selection of method.**
(DOCX)

**S2 File. Total number of ambulance active.**
(DOCX)

**S3 File. Distance and time analysis.**
(DOCX)

**S1 Data.**
(ZIP)

## Acknowledgments

The authors are grateful to the Emergency Medical Services, Department of Health and Family Welfare, Tamil Nadu, India, for providing the data for this study. The authors would also like to thank Mr. M Selvakumar, State Head of Operations, Tamil Nadu EMRI Green Health Services, for his assistance.

The authors would additionally like to thank Mr. Shahul Hameed, Inspector of Police, Technical Services, for making significant contributions to the overall presentation of the study, including the graphs. Importantly, the authors wish to express their gratitude to Mr Venkateswaran for his secretarial assistance in the preparation of the manuscript.

## Author Contributions

**Conceptualization:** Kandaswamy Paramasivan.

**Data curation:** Ashwin Prakash, Sarthak Gupta, Bhairav Phukan.

**Formal analysis:** Kandaswamy Paramasivan, Ashwin Prakash, Sarthak Gupta, Pavithra M.R., Balaji Venugopal.

**Investigation:** Kandaswamy Paramasivan, Ashwin Prakash, Pavithra M.R., Balaji Venugopal.

**Methodology:** Kandaswamy Paramasivan, Sarthak Gupta, Bhairav Phukan.

**Resources:** Kandaswamy Paramasivan.

**Validation:** Ashwin Prakash, Sarthak Gupta, Bhairav Phukan.

**Visualization:** Ashwin Prakash, Pavithra M.R., Balaji Venugopal.

**Writing – original draft:** Kandaswamy Paramasivan.

**Writing – review & editing:** Kandaswamy Paramasivan, Ashwin Prakash, Sarthak Gupta, Bhairav Phukan, Pavithra M.R., Balaji Venugopal.

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
