## [Decision Letter · Decision Letter 0]

2 Aug 2023

PONE-D-23-02384Resilience of hospital and allied infrastructure during pandemic and post pandemic periods for maternal health care of pregnant women and infants in Tamil Nadu, India - A counterfactual analysisPLOS ONE

Dear Dr. Paramasivan,

Thank you for submitting your manuscript to PLOS ONE. After careful consideration, we feel that it has merit but does not fully meet PLOS ONE’s publication criteria as it currently stands. Therefore, we invite you to submit a revised version of the manuscript that addresses the points raised during the review process.

We look forward to receiving your revised manuscript.

Kind regards,

Anil Gumber, Ph.D.

Academic Editor

PLOS ONE

Additional Editor Comments:

Could you please revise your paper in accordance to reviewers comments

Reviewers' comments:

Reviewer's Responses to Questions

**Comments to the Author**

1. Is the manuscript technically sound, and do the data support the conclusions?

Reviewer #1: Partly

Reviewer #2: Yes

2. Has the statistical analysis been performed appropriately and rigorously? 

Reviewer #1: Yes

Reviewer #2: Yes

3. Have the authors made all data underlying the findings in their manuscript fully available?

Reviewer #1: No

Reviewer #2: Yes

4. Is the manuscript presented in an intelligible fashion and written in standard English?

Reviewer #1: Yes

Reviewer #2: Yes

5. Review Comments to the Author

Reviewer #1: The manuscript “Resilience of hospital and allied infrastructure during pandemic and post pandemic

periods for maternal health care of pregnant women and infants in Tamil Nadu, India -

A counterfactual analysis” is a well-thought study demonstrating important learning and health systems inputs based on analysing secondary data analysis. The authors should consider the following issues.

General comments:

Although the authors followed the recommended structure of the journal there were several repetitions of methods-results-and discussion text. The method should clearly specify the study design, indicators explored and their definition and source of data, and the statistical analysis.

Discussion of cooperative advantages of several statical modelling is albeit important and justifies the strength, however, could be moved to an appendix. Only the detailed description of the model, its contraction and different variables/adjustments considered should be included in the main paper.

Cut-offs for categorising the level of effect size should be specified in the statistical analysis section.

All background and methods text and contextual descriptions that are not directly coming from the data analysis for this paper should be moved to the background and methods to avoid repetitions. Authors’ interpretation of data based on contextual knowledge should be moved to the discussion.

Please correct the typos and duplicate text where relevant.

Specific comment:

The MMR estimates provided in lines 116-118 should be revisited, should it be 113 per 100,000 live births (not per million)? Please correct others also.

On page 7, the authors specified the ambulance service for the routine ANC (without any emergency) is provided to high-risk women only. This should be clarified in the abstract (perhaps in results and discussion also) that these are not all pregnant women but only the high-risk groups.

May consider dropping structure description lines 160-166 when an IMRD structure is followed.

Line 185: Please specify the source of demographic data.

Did the authors consider the seasonality of pregnancy/births (if there is any)? How did they adjust that in the model to tease the seasonal effect of the higher number of pregnancy/ANC and birth care-seeking during certain months of the year?

Line 218: March 22, 2021, is a typo?

Line 241: correct repetition

Line 359: Why did the authors consider only three months for the validation data span? How would this validate the seasonal and other natural variations that might be happening in other months of the year? Why not use one full year for validation?

Lines 391-407 are not from the results from the data used for this manuscript and should be moved to background/methods

Line 409: methods of model accuracy testing and relevant cut-offs should be clarified in methods.

It appears that calls for ANC have increased after increasing ambulance but does not necessarily establishes a causal association. If the number of pregnancies remains and other things remain constant, has the selection/screening criteria or process to identify the high-risk pregnant women who are covered by CAS been changed any time after the onset of the pandemic? This might have resulted in several non-high-risk women who were not covered by CAS before started availing the service during and after the COVID-19 waves.

The distance analysis also needs to be revisited considering the case mix (type of clients/reasons). The case mix could be different before and post covid could be different and spread randomly whereas during the covid many cases could be clustered which might impact the average distance travelled and partly explain the lower distance travelled after the period new ambulances were introduced.

Figures 7: need to explain better the sudden jumps of NIFT pregnancy calls after April 2022 whereas the number of ambulances was increased since September 2020 (specified in line 463).

Thank you.

Reviewer #2: Summary:

The manuscript under review provides a comprehensive exploration of the COVID-19 pandemic's influence on antenatal care (ANC) and emergency delivery services within Tamil Nadu, India. It spans over three pandemic waves from 2020 to 2022. A rigorous counterfactual analysis aided by an auto regressive neural network (AR-Net) was employed by the authors to anticipate the daily volume of ambulance service calls (CAS), which they divided into three categories. The research findings suggest that the pandemic didn't detrimentally impact emergency or scheduled prenatal care. Quite the contrary, an appreciable increase in both the number of emergency-related IFT calls and prenatal care related CAS was detected during the specified periods. This signifies a robust response from the healthcare sector.

Major Comments:

1. Relevance and Originality: This research holds significant relevance in the contemporary global landscape and expands the existing body of literature with its unique employment of counterfactual analysis and neural network modeling to assess the pandemic's impact on healthcare services.

2. Methodology: The research methodology is well-defined and aligns seamlessly with the study's objectives. The integration of AR-Net into the predictive model and subsequent data segmentation deepens the study, revealing novel applications of machine learning in the realm of healthcare data analysis.

3. Results and Conclusions: The authors have succinctly presented their findings and supported them with the collected data. The paper grants valuable insights into the resilience and adaptability of healthcare services in Tamil Nadu throughout the pandemic waves. Its conclusion advocating for the extension of ambulance services and heightened awareness carries substantial implications for healthcare policy and pandemic preparedness.

Minor Comments:

1. Writing and Structure: The manuscript maintains a high degree of professionalism and clarity in its language. The logical structure aids the reader in understanding the study's aims, methodologies, and outcomes.

2. Presentation of Findings: Although the findings are logically and clearly presented, there could be scope for enhancing the visual representation or tabulation of the data, which would facilitate easy comparison between actual and predicted results.

3. Literature Citations: While the paper draws on a satisfactory range of relevant literature, it could gain further depth by referencing studies with similar methodologies. Moreover, I would suggest considering the inclusion of Ram & Sornette's study (2021) [1] on the impact of governmental interventions on epidemic progression and workplace activity during the COVID-19 outbreak. This could provide additional support and context to the discussion.

Recommendation:

This manuscript makes a valuable contribution to understanding the global health crisis's impact on healthcare services, with a particular emphasis on maternal care. The authors' innovative approach of utilizing counterfactual analysis and AR-Net modeling warrants commendation. The implementation of the study, coupled with the presentation of its findings, is well-executed. The insights gathered carry relevance for policy-making and future research.

Hence, I recommend the paper's acceptance in its current form, albeit with minor revisions to augment data representation and further enrich the literature review.

[1] Ram, S. K., & Sornette, D. (2021). Impact of Governmental interventions on epidemic progression and workplace activity during the COVID-19 outbreak. Scientific Reports, 11(1), 21939.

6. PLOS authors have the option to publish the peer review history of their article (what does this mean?). If published, this will include your full peer review and any attached files.

Reviewer #1: No

Reviewer #2: No

<quillbot-extension-portal></quillbot-extension-portal>

---

## [Author Response · Author response to Decision Letter 0]

16 Aug 2023

REVIEWER #1

General comments(GC)

We thank the Academic Editor and the reviewers for thoroughly reading the paper and coming up with excellent suggestions, detailed comments, and constructive remarks to improve the quality of the article. We have meticulously analysed all the comments and have appropriately addressed the reviewers' concerns.

GC # 1 Although the authors followed the recommended structure of the journal there were several repetitions of methods-results-and discussion text. The method should clearly specify the study design, indicators explored and their definition and source of data, and the statistical analysis.

We thank the reviewer for pointing out the repetitions, and appropriate changes have been incorporated. A new subsection titled ‘Study Design’ has been added in the Method section. The unnecessary repetitions have been removed in the method, results, and discussion sections.

GC# 2 Discussion of cooperative advantages of several statical modelling is albeit important and justifies the strength, however, could be moved to an appendix. Only the detailed description of the model, its contraction and different variables/adjustments considered should be included in the main paper.

In response to the feedback provided by the reviewer, we have made revisions to the method section. The retained information consists solely of a concise depiction of the model and the variables being examined. The remaining sections have been relocated to an appendix. It is now placed as Supplementary Information S-1: Background – Selection of Method.

GC# 3 Cut-offs for categorising the level of effect size should be specified in the statistical analysis section.

Acknowledging the reviewer's remark, we have added a brief description of the ranges for the effect size, including the cut-offs in the revised manuscript. Please see the last paragraph of Method section. 

GC# 4 All background and methods text and contextual descriptions that are not directly coming from the data analysis for this paper should be moved to the background and methods to avoid repetitions. Authors’ interpretation of data based on contextual knowledge should be moved to the discussion.

Both of the remarks above have been addressed. The revised article has undergone reorganisation, with the relocation of specific paragraphs from the results section to the method section. Likewise, the interpretation of the data and results has been moved to the discussion section.

GC# 5 Please correct the typos and duplicate text where relevant.

Per the referee's suggestion, the paper has been edited and proofread to remove repetitions and correct typos.

GC# Specific comment:

SC# 1 The MMR estimates provided in lines 116-118 should be revisited, should it be 113 per 100,000 live births (not per million)? Please correct others also.

We once again thank the reviewer for pointing out this mistake. We have made the appropriate corrections in the revised manuscript.

SC# 2 On page 7, the authors specified the ambulance service for the routine ANC (without any emergency) is provided to high-risk women only. This should be clarified in the abstract (perhaps in results and discussion also) that these are not all pregnant women but only the high-risk groups.

The government programme includes all pregnant women rather than exclusively targeting those at high risk. The government initiative is known as SUMAN is designed to offer cost-free antenatal care services to pregnant women who attend primary health centres. In the revised manuscript, it is appropriately clarified.

SC# 3 May consider dropping structure description lines 160-166 when an IMRD structure is followed.

We agree with the reviewer's suggestion. As the article is structured in IMRD format, lines 160–166 have been removed.

SC# 4 Line 185: Please specify the source of demographic data.

While receiving the call, the call taker in the 108 Ambulance Control Room seeks certain demographic information from the caller, such as age, gender, employment, community, and socioeconomic status. However, it is only optional for the caller to divulge; hence, demographic information is not fully available except for age and gender, which must be furnished. In the revised manuscript, we have made appropriate changes.

SC# 5 Did the authors consider the seasonality of pregnancy/births (if there is any)? How did they adjust that in the model to tease the seasonal effect of the higher number of pregnancy/ANC and birth care-seeking during certain months of the year?

The authors used the Dickey-Fuller test to determine the data's stationarity, revealing no seasonality in the pregnancy time series data. Despite the possibility of the test failing to capture seasonality, the study remains robust as the neural network's hybrid model (Neural Prophet) incorporates the seasonality inherent in the time series data when making predictions. In order to address the issue of seasonality, N-Prophet adopts the approach of seeing the time series as a periodic and continuous function, which is mathematically represented as a Fourier series in equation 3, of the revised paper.

SC# 6 Line 218: March 22, 2021, is a typo?

We thank the reviewer for pointing out this typo. It has now been corrected as March 22, 2022.

SC# 7 Line 241: correct repetition

We again thank the reviewer for pointing out the repetition, which has been corrected in the updated manuscript.

SC# 8 Line 359: Why did the authors consider only three months for the validation data span? How would this validate the seasonal and other natural variations that might be happening in other months of the year? Why not use one full year for validation?

Considering that the data set was not sufficiently large for tuning many parameters, we trained the model with four years of data and tested the model with unseen data in a validation period of three months. Ideally, one full year would have been appropriate. However, due to the limitations of the data from the control centre, we could not validate it for the complete one-year period. We have used the same three-month period for validation in some of our articles that have been published in PLoS ONE, Nature: Humanities and Social Sciences, Policing and Society, and Heliyon.

• Paramasivan, K., et al. (2022). Crime registration and distress calls during COVID-19: two sides of the coin Policing and Society, 1–22. https://doi.org/10.1080/10439463.2021.2023526

• Paramasivan, K., et al. (2022). Relationship between mobility and road traffic injuries during the COVID-19 pandemic—The role of attendant factors. PloS one, 17(5), e0268190. https://doi.org/10.1371/journal.pone.0268190

• Paramasivan, K., et al. (2022). Empirical evidence of the impact of mobility on property crimes during the first two waves of the COVID-19 pandemic. NATURE-Humanities and Social Sciences Communications. 9, 373 (2022). https://doi.org/10.1057/s41599-022-01393-0

• Paramasivan K. et al. (2023) Prolonged school closure during the pandemic time in successive waves of COVID-19- vulnerability of children to sexual abuses - A case study in Tamil Nadu, India Heliyon. 2023 Jul;9(7):e17865. doi:10.1016/j.heliyon.2023.e17865. PMID: 37456023.

SC# 9 Lines 391-407 are not from the results from the data used for this manuscript and should be moved to background/methods

Agreeing with the reviewers comments, these lines have been moved out. 

SC# 10 Line 409: methods of model accuracy testing and relevant cut-offs should be clarified in methods.

We express our gratitude to the reviewer for recommending the inclusion of ranges to represent the model's accuracy or error levels. The method section in the revised manuscript includes the relevant thresholds for acceptable error levels in the prediction model. 

SC# 11 It appears that calls for ANC have increased after increasing ambulance but does not necessarily establishes a causal association. If the number of pregnancies remains and other things remain constant, has the selection/screening criteria or process to identify the high-risk pregnant women who are covered by CAS been changed any time after the onset of the pandemic? This might have resulted in several non-high-risk women who were not covered by CAS before started availing the service during and after the COVID-19 waves.

It is worth noting that, in addition to the rise in ambulance availability, the Government undertook a strident campaign to promote ambulance services in both urban and rural regions. Moreover, it is possible that pregnant women were more inclined to utilise the healthcare services provided by government hospitals following their close scrutiny of the government healthcare sector's response during the pandemic. It is unambiguously clear that SUMAN, the Government initiative to provide maternal health care for pregnant women, is open to any woman who has visited a primary health care centre. Hence, the rise in ANC demand can be linked to the expansion of the ambulance fleet, the growing awareness and utilisation of ambulance services, and the increasing voluntary preference of pregnant women to access these services. Importantly, there was no change in the norms for admitting pregnant women to the ANC scheme during the pandemic period.

SC# 12 The distance analysis also needs to be revisited considering the case mix (type of clients/reasons). The case mix could be different before and post covid could be different and spread randomly whereas during the covid many cases could be clustered which might impact the average distance travelled and partly explain the lower distance travelled after the period new ambulances were introduced.

We acknowledge the valid observation of the reviewer to revisit the distance and time analysis. There is a detailed explanation along with the results of the analysis in the supplementary section in S-2 Distance and Time Analysis. However it is reproduced here for quick reference

The distance and duration study done for the total calls, irrespective of the type of medical emergency, slightly differs from the sub-group. For instance, both the distance travelled (measured in kilometres [km]) by the ambulance and the time consumed (measured in minutes) in respect of those calling for emergencies such as acute abdomen pain, pregnancy-related concerns, and COVID-19 illnesses were higher than the norm. On the contrary, callers who needed ambulance service for treatment as trauma victims on account of road crashes and cardio-vascular disorders had speedier transportation to the hospital and covered a lesser distance than average. Table S-2 displays those people with crises who had faster transportation and a shorter distance to go, whereas Table S-3 shows the people with emergencies who had a greater distance and a longer time.

SC# 13 Figures 7: need to explain better the sudden jumps of NIFT pregnancy calls after April 2022 whereas the number of ambulances was increased since September 2020 (specified in line 463).

The data pertaining to the growth in the number of ambulances in service from April 2018 to November 2022 has been compiled and presented as S 2 in the supplementary section. In February 2020, the fleet size was recorded as 1237. Subsequently, in March 2022, an increase of more than 100 ambulances occurred, resulting in a total inventory of 1353. The aforementioned rise is also a contributing factor to the escalation in demand for ambulance services catering to pregnant women. Nevertheless, it is worth noting that there were additional significant factors that contributed to the success of the ambulance services. These factors include a vigorous government campaign aimed at promoting the utilisation of ambulance services, as well as the voluntary inclination of pregnant women to choose government-provided antenatal care (ANC) services. This decision was likely influenced by the favourable response of government hospitals to effectively managing the pandemic.

REVIEWER#2

Major Comments:

1. Relevance and Originality: This research holds significant relevance in the contemporary global landscape and expands the existing body of literature with its unique employment of counterfactual analysis and neural network modeling to assess the pandemic's impact on healthcare services.

We thank the reviewer for the comment.

2. Methodology: The research methodology is well-defined and aligns seamlessly with the study's objectives. The integration of AR-Net into the predictive model and subsequent data segmentation deepens the study, revealing novel applications of machine learning in the realm of healthcare data analysis.

We are indeed thankful for the appreciation of the methodology used.

3. Results and Conclusions: The authors have succinctly presented their findings and supported them with the collected data. The paper grants valuable insights into the resilience and adaptability of healthcare services in Tamil Nadu throughout the pandemic waves. Its conclusion advocating for the extension of ambulance services and heightened awareness carries substantial implications for healthcare policy and pandemic preparedness.

We express our gratitude to the reviewer for their meaningful comment.

Minor Comments:

1. Writing and Structure: The manuscript maintains a high degree of professionalism and clarity in its language. The logical structure aids the reader in understanding the study's aims, methodologies, and outcomes.

We thank the reviewer for the positive feedback.

2. Presentation of Findings: Although the findings are logically and clearly presented, there could be scope for enhancing the visual representation or tabulation of the data, which would facilitate easy comparison between actual and predicted results.

In response to the aforementioned comment made by the reviewer, we have incorporated a figure with six panels that visually depict the actual /observed and projected/predicted call volumes for the primary call categories, namely Total, Vehicular Trauma, and Pregnancy-related calls. Please see Fig 10. 

3. Literature Citations: While the paper draws on a satisfactory range of relevant literature, it could gain further depth by referencing studies with similar methodologies. Moreover, I would suggest considering the inclusion of Ram & Sornette's study (2021) [1] on the impact of governmental interventions on epidemic progression and workplace activity during the COVID-19 outbreak. This could provide additional support and context to the discussion.

In accordance with the recommendation of the reviewer, we have included this significant citation in our amended work.

Recommendation:

This manuscript makes a valuable contribution to understanding the global health crisis's impact on healthcare services, with a particular emphasis on maternal care. The authors' innovative approach of utilizing counterfactual analysis and AR-Net modeling warrants commendation. The implementation of the study, coupled with the presentation of its findings, is well-executed. The insights gathered carry relevance for policy-making and future research.

Hence, I recommend the paper's acceptance in its current form, albeit with minor revisions, to augment data representation and further enrich the literature review.

We thank the reviewer for recommending the article for publication.

---

## [Editor Report · Decision Letter 1]

5 Sep 2023

Resilience of hospital and allied infrastructure during pandemic and post pandemic periods for maternal health care of pregnant women and infants in Tamil Nadu, India - A counterfactual analysis

PONE-D-23-02384R1

Dear Dr. Paramasivan,

We’re pleased to inform you that your manuscript has been judged scientifically suitable for publication and will be formally accepted for publication once it meets all outstanding technical requirements.

Kind regards,

Anil Gumber, Ph.D.

Academic Editor

PLOS ONE

Additional Editor Comments (optional):

Thanks for addressing reviewers comments.
---

## [Editor Report · Acceptance letter]

8 Sep 2023

PONE-D-23-02384R1 

Resilience of hospital and allied infrastructure during pandemic and post pandemic periods for maternal health care of pregnant women and infants in Tamil Nadu, India - A counterfactual analysis 

Dear Dr. Paramasivan:

I'm pleased to inform you that your manuscript has been deemed suitable for publication in PLOS ONE. Congratulations! Your manuscript is now with our production department. 

Kind regards, 

on behalf of

Dr. Anil Gumber 

Academic Editor

PLOS ONE